# ATTENTION MISALIGNMENT ATTACKS: TARGETING CROSS-MODAL ATTENTION IN MULTIMODAL LARGE LANGUAGE MODELS FOR ADVERSARIAL EXAMPLES

## ABSTRACT

Multimodal large language models (MLLMs) have achieved impressive performance across a wide range of multimodal understanding tasks. However, their growing deployment raises concerns about robustness under adversarial conditions. Existing adversarial attacks on MLLMs predominantly focus on disrupting the global semantic alignment between image and text by optimizing over joint embeddings or globally aggregated image/text token representations. We observe that such methods often fail to generate effective adversarial examples for fine-grained tasks such as VQA (Visual Question Answering), especially when the questions aim at detailed understanding of particular regions in the image, which requires precise alignment between image regions and textual answers for MLLMs. To address this, we propose Attention Misalignment Attack (AMA), a novel plug-and-play attack method that is highly compatible with existing attack objectives—it can be easily integrated by combining its attention misalignment loss with other attack losses. AMA operates by extracting attention maps from each decoding step of the MLLM and optimizing the divergence between target and adversarial attention patterns, guided by semantic similarity. This forces the model to attend to irrelevant regions, effectively misguiding its answer generation process even towards fine-grained questions. To improve efficiency, we further introduce FastAMA, a lightweight variant that avoids autoregressive decoding and instead uses a single forward pass to extract self-attention from the input tokens. Experiments show that our method significantly enhances the performance of existing attack methods across multiple tasks, especially on the more challenging instances within VQA datasets.

## 1 INTRODUCTION

Multimodal large language models (MLLMs), which integrate visual and textual information into unified architectures, have demonstrated remarkable capabilities in a wide range of vision-language tasks, including visual question answering (VQA) Antol et al. (2015), image captioning, and visual dialogue. By leveraging pre-trained vision encoders (e.g., CLIP Radford et al. (2021), ViT Dosovitskiy et al. (2021)) and powerful large language models (e.g., LLaMA Touvron et al. (2023), GPT Team (2024)), MLLMs are able to perform complex understanding grounded in visual content, making them a promising paradigm for real-world AI applications.

Despite these advances, recent studies Cui et al. (2024); Zhao et al. (2023); Dong et al. (2023) have raised concerns about the robustness and security of MLLMs under adversarial conditions. While extensive research has explored adversarial attacks on unimodal models in vision and NLP Goodfellow et al. (2014); Dong et al. (2019); Chen et al. (2023); Ma et al. (2023), the multimodal setting introduces new challenges: attacks must simultaneously affect both visual and textual modalities and often interact with complex cross-modal alignment mechanisms. In particular, MLLMs rely heavily on cross-modal attention, which governs how the language model attends to visual features during generation. However, most existing adversarial attacks for MLLMs focus on global semantic representations—such as joint embeddings or overall image/text token matching—while largely ignoring the internal mechanisms that guide fine-grained visual understanding.

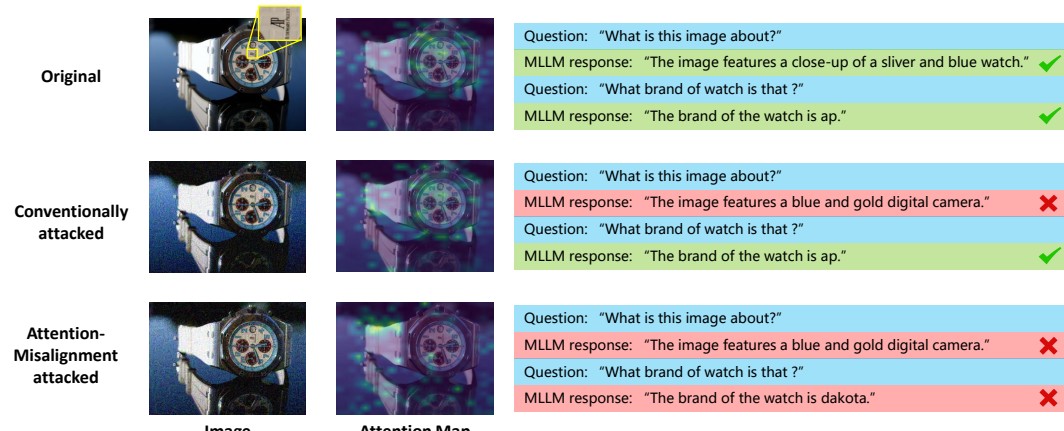

Figure 1: The adversarial example, attention map in MLLM, and response from MLLM after performing conventional attack and out Attention-Misalignment Attack.

For instance, we observe that the VQA performance of most existing attacks are tested on relatively simple VQA datasets like VQA-v2 Goyal et al. (2017), where questions are general and can often be answered correctly via global image understanding. In contrast, datasets such as TextVQA Singh et al. (2019) contain more specific and localized questions, such as reading text within an image or counting detailed elements, which demand precise image-to-text alignment from the model. Attacking these models under such settings is inherently more difficult. Figure 1 presents an example from the TextVQA dataset to highlight the limitations of conventional attack methods, where existing methods successfully manipulate MLLMs to misinterpret the overall content of an image, but fail to affect the model's responses to more detailed questions. We observe that even after a "successful attack", the average attention maps extracted from LLM decoder between the visual tokens of the adversarial and original images remain largely similar when the decoder generates each output token. Since these attention distributions are critical for guiding the model's visual understanding, the lack of direct manipulation in this stage could be a key factor behind the failure of conventional methods to mislead MLLMs on detailed queries.

To address this limitation, we propose Attention Misalignment Attack (AMA), a novel adversarial framework that directly targets the cross-modal attention maps in the MLLM decoder. At each decoding step, we weight attention maps by the semantic relevance of generated tokens to the ground truth, and optimizes adversarial perturbations to minimize the divergence between the targeted and perturbed attention maps, focusing specifically on attention directed toward image tokens. As shown in Figure 1, the attention maps exhibit notable shifts after AMA being applied, which misguides the model's visual understanding and leads to incorrect answer generation. We further develop **FastAMA**, an efficient variant that extracts final-layer self-attention maps from a single forward pass over the question and image tokens, achieving significant speedup with minimal performance loss. Finally, AMA/FastAMA is modular and can be seamlessly integrated with existing attacks an auxiliary objective alongside other loss functions, which enables our method to serve as a general enhancement for improving both the effectiveness and transferability of existing attack frameworks.

In our experiments, we evaluate the proposed method across widely used benchmarks in Image Classification, Image Captioning, and VQA. Additionally, we include evaluations on more challenging VQA datasets to further test the effectiveness of our method. The models under attack include commonly used open-source MLLMs such as LLaVA Liu et al. (2023), MiniGPT-4 Zhu et al. (2023), and InternVL Chen et al. (2024). Extensive experiments built upon three representative attack methods demonstrate that our proposed AMA consistently achieves significant improvements in attack success rates across different tasks, model architectures, and attack settings—most notably on the more difficult VQA benchmarks. Our contributions are summarized as follows:

- We propose a plug-and-play attack method named Attention Misalignment Attack (AMA), a novel framework that explicitly targets the attention maps between image tokens and decoder outputs, enabling effective disruption of the model's internal visual understanding.

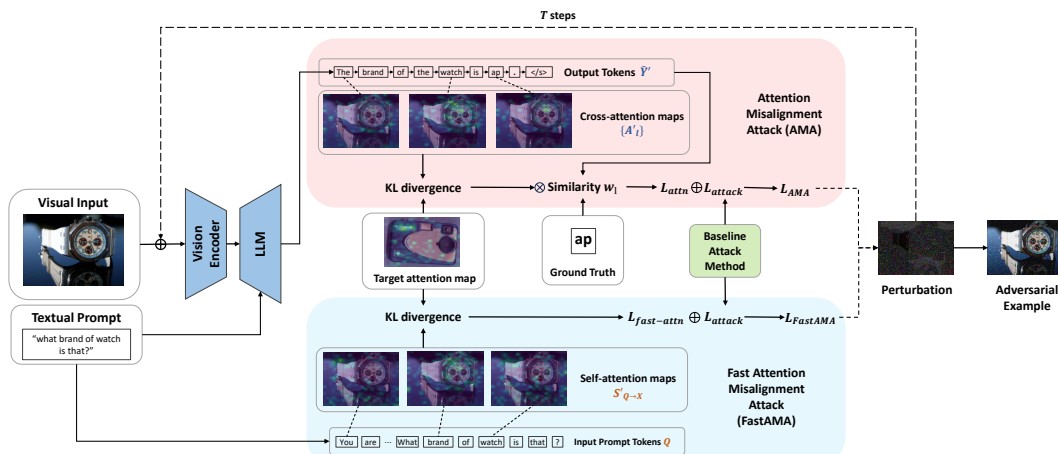

Figure 2: The overall pipeline of Attention Misalignment Attack(AMA) and Fast Attention Misalignment Attack (FastAMA).

- To further improve efficiency, we develop FastAMA, a lightweight variant of AMA with significantly reduced computational cost and memory usage, while maintaining comparable effectiveness.
- Our method is **plug-and-play** and model-agnostic, requiring no modification to the target model or base attack pipeline, and can be flexibly applied to various attack methods.
- Extensive experiments show that our method achieves consistent and significant gains across various tasks, model architectures, and attack settings.

## 2 RELATED WORKS

### 2.1 MULTIMODAL LARGE LANGUAGE MODELS

Recent advances in Multimodal Large Language Models (MLLMs) have led to significant breakthroughs in visual-language understanding tasks. MLLMs generally adopt a two-tower paradigm: a vision tower (e.g. CLIP Radford et al. (2021)/EVA-CLIP Sun et al. (2023)) encodes images and a language tower (e.g. Vicuna Chiang et al. (2023)/LLaMA Touvron et al. (2023)) processes text. To bridge the gap between text and other modalities, early MLLMs freeze both vision and language tower. BLIP-2 Li et al. (2023) and InstructBLIP Dai et al. (2023) train a 12-layer Querying Transformer (Q-Former) between 2 towers that learns to extract "query" tokens from image features, while MiniGPT-4 Zhu et al. (2023) aligns a frozen ViT image encoder with a Vicuna LLM via a single linear projection layer. LLaVA Liu et al. (2023) takes a further step by connecting CLIP's visual encoder to Vicuna and fine-tuning end-to-end. Newer work like Qwen-VL Bai et al. (2025) and InternVL Chen et al. (2024), however, scales the vision tower to billions of parameters and aligns it with LLMs via a staged training pipeline, resulting in a unified model that yields state-of-the-art on dozens of multimodal benchmarks.

### 2.2 ADVERSARIAL ROBUSTNESS ON MLLMS

At the same time, MLLMs inherit vulnerabilities from both vision and language domains. In pure vision domain, Zhao et al. Zhao et al. (2023) craft targeted adversarial examples against pretrained models and transfer them to MLLMs. Cui et al. Cui et al. (2024) conduct a systematic study of white-box image attacks on MLLMs, finding that purely visual perturbations can break classification and captioning tasks. Recently researchers have focused on hybrid attacks. CroPA Luo et al. (2024) updates the visual adversarial perturbation with learnable prompts to improve the transferability of adversarial examples across prompts. Kim et al. Kim et al. (2024) utilize the attention mechanism in vision encoder to generate UAP that deceives MLLMs across both image and text input. DynVLA Gu et al. (2025) injects dynamic perturbations into the vision-language connector to

enhance generalization across diverse vision-language alignment of different models. Despite these advances, most existing approaches predominantly target the model's global understanding of the image, leaving their fine-grained visual comprehension largely intact. Consequently, such methods often fail in scenarios where precise understanding over detailed visual content is required.

## 3 METHOD

### 3.1 PRELIMINARY

Given a MLLM input instance composed of an input image $X$, a textual prompt $Q$, and a ground truth answer $Y$, the MLLM $f$ generates an answer $\hat{Y} = f(X, Q)$ token by token via an autoregressive decoding process.

In the untargeted setting, the goal is to generate an adversarial image $X' = X + \delta$ that causes the model to produce an output $\hat{Y}' = f(X', Q)$ that is semantically different from the ground-truth answer $Y$. In the targeted setting, the attacker is provided with a predefined target output $Y^*$, and aims to generate an adversarial image $X'$ such that the model's prediction matches $Y^*$, i.e., $\hat{Y}' = f(X', Q) = Y^*$.

In both cases, the adversarial image $X'$ must remain within a small perturbation budget $\epsilon$, typically enforced under the $L_\infty$ norm constraint: $\|X' - X\|_\infty \leq \epsilon$. This ensures that the adversarial perturbations remain imperceptible to humans while being effective against the MLLM.

### 3.2 ATTENTION MISALIGNMENT ATTACK

While the aforementioned objectives form the basis of conventional adversarial attacks on MLLMs, they often focus solely on altering the model's final output while overlooking how the model internally attends to visual content during the generation process. To address this gap, we propose Attention Misalignment Attack (AMA), a novel targeted adversarial attack framework for Multimodal Large Language Models (MLLMs), which focuses on disrupting the alignment between generated textual tokens and visual content in a fine-grained manner. In contrast to prior methods that target global embeddings or token-level representations, AMA exploits the attention patterns in the MLLM decoder to craft input perturbations that directly distort the model's visual understanding during answer generation.

#### 3.2.1 ATTENTION MISALIGNMENT LOSS

Let the model decode a sequence of tokens $\hat{Y}' = \{y_1, y_2, ..., y_L\}$, where $L$ is the length of the output. During each decoding step $l$, the model attends over a set of visual tokens with a learned attention distribution $\mathbf{A}_l \in \mathbb{R}^N$, where $N$ is the number of visual tokens.

Given the targeted image $X^*$ and the adversarial image $X'$, we denote the corresponding attention maps as $\mathbf{A}_l$ and $\mathbf{A}'_l$, respectively. We define the attention misalignment loss as:

$$\mathcal{L}_{\text{attn}} = \sum_{l=1}^{L} w_l \cdot D(\mathbf{A}_l^*, \mathbf{A}'_l)$$

where $D(\cdot, \cdot)$ is a distance function between two attention distributions, here we use KL divergence. $w_t$ is a token importance weight that reflects the semantic contribution of each token to the answer.

This loss encourages the adversarial image to lead the model to attend differently during decoding, misguiding its internal understanding.

#### 3.2.2 TOKEN WEIGHTING

Not all output tokens are equally important. To reflect this, AMA assigns dynamic weights $w_t$ to each token. By default, we compute token importance using edit distance between the model's current output and the ground truth:

$$w_l = \text{Sim}(y_l, Y)$$

tokens that similar to ground truth are given higher weights, indicating a need to shift attention for those steps. The Alternative weighting strategies are discussed in experiments section.

### 3.2.3 INTEGRATION WITH EXISTING ATTACKS

AMA is designed to be plug-and-play, meaning it can be directly added to existing attack methods. Given an original adversarial loss $\mathcal{L}_{\text{attack}}$ defined in the base attack method, we define the total loss of AMA as:

$$\mathcal{L}_{\text{AMA}} = \mathcal{L}_{\text{attack}} + \lambda \cdot \mathcal{L}_{\text{attn}}$$

where $\lambda$ balances the contribution of the attention misalignment loss.

To further stabilize the attack process, we explore dynamic adjustment of $\lambda$ using uncertainty weighting Cipolla et al. (2018), which adapts the scale of each loss based on its empirical variance during the attack optimization process.

Finally, we perform gradient-based optimization over the image perturbation $\delta$ under $\ell_\infty$ norm constraints to minimize $\mathcal{L}_{\text{AMA}}$.

### 3.3 FAST AMA

While AMA effectively manipulates cross-modal attention during generation, it requires extracting attention maps at every decoding step, which leads to substantial computational overhead, especially for long answer sequences and multi-step adversarial optimization.

To address this, we propose FastAMA, a computationally efficient variant that avoids repeated autoregressive generation during attack. Instead of relying on decoder attention maps from each generated token, FastAMA computes a single forward pass through the MLLM with a fixed input composed of the question text tokens and image tokens. It then extracts the self-attention map $\mathbf{S}$ from the final layer of the decoder Transformer.

From $\mathbf{S}$, we isolate the submatrix corresponding to attention from question tokens to image tokens, denoted as $\mathbf{S}_{Q \to X}$. This submatrix captures how much the model attends to image regions when processing the question, and serves as a proxy for the model's understanding.

Similarly, We define the Fast Attention Misalignment Loss by:

$$\mathcal{L}_{\text{fast-attn}} = D(\mathbf{S}^*_{Q \to X}, \mathbf{S}'_{Q \to X})$$

where $\mathbf{S}^*_{Q \to X}, \mathbf{S}'_{Q \to X}$ denotes the targeted attention map and the attention map towards adversarial image, respectively. This avoids token-wise generation and results in a drastic reduction in attack runtime.

Finally, we combine $\mathcal{L}_{\text{fast-attn}}$ with the same task-specific adversarial objective used in AMA to form the full FastAMA loss:

$$\mathcal{L}_{\text{FastAMA}} = \mathcal{L}_{\text{attack}} + \lambda \cdot \mathcal{L}_{\text{fast-attn}}$$

The whole attack process of AMA and FastAMA is described in Algorithm 1.

## 4 EXPERIMENTS

### 4.1 MLLMS AND DATASETS

To ensure the fairness of our comparisons, we adhere to the experimental settings adopted in prior work and conduct evaluations on MiniGPT-4 Zhu et al. (2023) and LLaVA Liu et al. (2023). For dataset selection, MS-COCO Lin et al. (2014) is utilized for both the image classification and image captioning tasks. In the evaluation of VQA, we further differentiate between two levels of task difficulty. The first setting, denoted as VQA-Easy, follows the conventional benchmark used in prior works and is based on the VQA-v2 Goyal et al. (2017) dataset. The second, referred to as VQA-Hard, is constructed using the TextVQA Singh et al. (2019) dataset, which requires more fine-grained visual understanding and imposes greater challenges for both model inference and adversarial attack. To ensure that the model capabilities are well-aligned with the increased complexity of the VQA-Hard setting, we further include InternVL2.5 Chen et al. (2025)—a state-of-the-art

---

**Algorithm 1:** Attention Misalignment Attack (AMA) and FastAMA

---

**Input:** Image $X$, question $Q$, MLLM $f$, ground truth $Y$, steps $T$, weight $\lambda$, attack step size $\alpha$

**Output:** Adversarial image $X'$

1  Initialize $X' \leftarrow X$
2  **for** $t = 1$ *to* $T$ **do**
3      Compute original adversarial loss $\mathcal{L}_{\text{attack}}$
4      **if** *Using AMA* **then**
5          Generate answer $\hat{Y}'$ token-by-token from $f(X', Q)$
6          $\mathcal{L}_{\text{attn}} \leftarrow 0$
7          **for** *each token $y_l$ in $\hat{Y}'$* **do**
8              Weight $w_l \leftarrow \text{Sim}(y_l, Y)$
9              Extract attention $\mathbf{A}_l$ at decoder layer
10             $\mathcal{L}_{\text{attn}} \mathrel{+}= w_l \cdot D(\mathbf{A}_l^*, \mathbf{A}_l')$
11     **else**
12         Forward pass on $f(X', Q)$
13         Extract final-layer attention $\mathbf{S}_{Q \to X}$
14         $\mathcal{L}_{\text{attn}} \leftarrow D(\mathbf{S}_{Q \to X}^*, \mathbf{S}_{Q \to X}')$
15     $\mathcal{L}_{\text{(Fast)AMA}} \leftarrow \mathcal{L}_{\text{attack}} + \lambda \cdot \mathcal{L}_{\text{attn}}$
16     Update $X' \leftarrow \text{Clip}(X' - \alpha \cdot \nabla_{X'} \mathcal{L}_{\text{(Fast)AMA}})$
17 **return** $X'$

---

Table 1: Attack success rates (%) of the baseline MF-Attack, AMA-enhanced, and FastAMA-enhanced methods on four attack tasks (VQA-Hard, VQA-Easy, Classification, and Captioning) using three MLLMs: MiniGPT-4, LLaVA, and InternVL.

| Threat Model | Method | VQA-Hard | VQA-Easy | Classification | Captioning |
|---|---|---|---|---|---|
| MiniGPT-4 | Baseline | 31.2 | 66.7 | 50.3 | 15.4 |
| | +AMA | **50.0** (+18.8) | **83.2** (+16.5) | 56.1 (+5.8) | 19.1 (+3.7) |
| | +FastAMA | 45.6 (+14.4) | 77.9 (+11.2) | 53.5 (+3.2) | **21.0** (+5.6) |
| LLaVa | Baseline | 38.4 | 73.3 | 43.1 | 24.6 |
| | +AMA | **57.8** (+19.4) | **77.2** (+3.9) | 49.4 (+6.3) | **28.7** (+4.1) |
| | +FastAMA | 52.5 (+14.1) | 76.8 (+3.5) | **49.9** (+6.8) | 24.5 (-0.1) |
| InternVL | Baseline | 34.6 | 69.0 | 48.2 | 18.9 |
| | +AMA | **53.4** (+18.8) | 79.1 (+10.1) | 50.8 (+2.6) | **19.5** (+0.6) |
| | +FastAMA | 53.0 (+18.4) | **80.3** (+11.3) | **51.2** (+3.0) | 19.2 (+0.3) |

open-source MLLM with stronger vision-language understanding capacity—in our experimental comparisons.

## 4.2 IMPLEMENTATION DETAILS

To validate the general effectiveness of our proposed AMA and FastAMA across different attack paradigms, we conduct experiments based on three attack methods: MF-Attack Zhao et al. (2023), Attack-Bard Dong et al. (2023), and CroPA Luo et al. (2024). For each of these attacks, we retain all original settings and hyperparameters, except that we standardize the number of attack steps to $T = 100$ and fix the perturbation budget to $\epsilon = 16/255$. All experiments are conducted under the targeted attack setting for consistency and comparability. For Image Classification, Image Captioning, and VQA-Easy tasks, the textual target is uniformly set to the generic token "Unknown". For the VQA-Hard tasks, the target is chosen from the paired question-answer set, where the answers are concise and short. For AMA's token weighting computation, we measure the maximum similarity between each output token and a set of key content words extracted from the ground truth. Specifically, for VQA-Hard, the ground-truth answers themselves are treated as the keyword set; for Classification, Captioning, and VQA-Easy tasks, we use NLTK Bird & Loper (2004) to extract the key content words from the target answer sentence. All experiments are conducted on a single NVIDIA A800 GPU.

Table 2: Attack success rates (%) of AMA and FastAMA as plug-in modules for various attacks (MF-Attack, Attack-Bard, and CroPA) on four tasks, conducted on InternVL model.

| Attack Method | AMA Status | VQA-Hard | VQA-Easy | Classification | Captioning |
|---|---|---|---|---|---|
| | w/o AMA | 34.6 | 69.0 | 48.2 | 18.9 |
| MF-Attack | +AMA | **53.4** (+18.8) | 79.1 (+10.1) | 50.8 (+2.6) | **19.5** (+0.6) |
| | +FastAMA | 53.0 (+18.4) | **80.3** (+11.3) | **51.2** (+3.0) | 19.2 (+0.3) |
| | w/o AMA | 40.3 | 77.2 | 56.0 | 22.1 |
| Attack-Bard | +AMA | 55.5 (+15.2) | **89.8** (+12.6) | **61.0** (+5.0) | **27.7** (+5.6) |
| | +FastAMA | **57.3** (+17.0) | 89.5 (+12.3) | 59.4 (+3.4) | 25.8 (+3.7) |
| | w/o AMA | 45.5 | 82.3 | 60.4 | 35.5 |
| CroPA | +AMA | **66.6** (+21.1) | **87.5** (+5.2) | **65.4** (+5.0) | 38.4 (+2.9) |
| | +FastAMA | 66.1 (+20.6) | 83.5 (+1.2) | 63.2 (+2.8) | **38.6** (+3.1) |

## 4.3 PERFORMANCE ACROSS MODELS & DATASETS

As shown in Table 1, we conduct a comprehensive evaluation of the proposed AMA and FastAMA across four tasks and three representative MLLMs. In this context, the "Baseline" refers to the MF-Attack method, while "+AMA" and "+FastAMA" denote the results obtained by equipping MF-Attack with our AMA and FastAMA enhancements, respectively. From the results, it is evident that both AMA and FastAMA lead to substantial improvements in attack success rates across all models and tasks. Notably, on the VQA-Hard benchmark, the improvement reaches up to 50%, highlighting the effectiveness of our method in scenarios that demand fine-grained visual understanding.

While VQA-Easy, Classification, and Captioning—tasks commonly used in prior adversarial work—also benefit from the proposed techniques, the gains are particularly significant in VQA-Hard, where existing methods typically struggle. When comparing AMA and FastAMA directly, we observe that AMA tends to outperform FastAMA slightly in most cases. This supports our design intuition that assigning token-wise importance during gradient optimization enables more effective perturbation of the model's visual attention, especially in tasks requiring longer, descriptive answers.

Notably, on InternVL, FastAMA performs on par with, or occasionally even better than, AMA. We attribute this to the inherent strength of InternVL, which is pretrained extensively on diverse and challenging datasets. Unlike models such as MiniGPT-4 and LLaVA—which often generate verbose responses to VQA prompts (e.g., starting with "This image describes..."), where token importance modeling is crucial—InternVL tends to generate concise, accurate answers with just one or two tokens. In such settings, the benefit of token-level weighting is less pronounced, allowing the lighter-weight FastAMA to achieve comparable performance.

## 4.4 PLUG-AND-PLAY PERFORMANCE ON EXISTING METHODS

Similar to the previous section, to further validate the transferability and general effectiveness of our proposed AMA and FastAMA as plug-and-play modules, we extended our experiments beyond the original baseline (MF-Attack) to include two additional representative multimodal large model (MLLM) attack methods: Attack-Bard and CroPA. These experiments were conducted on the stronger InternVL model. As shown in Table 2, among the three attack methods, CroPA achieves the highest attack success rate, followed by Attack-Bard and then MF-Attack. Nevertheless, regardless of the underlying attack framework, integrating AMA or FastAMA consistently yields significant performance gains, especially on challenging tasks such as VQA-Hard. Furthermore, due to InternVL's advanced understanding capabilities, FastAMA performs comparably to AMA with minimal performance drop, demonstrating its efficiency-effectiveness trade-off. These findings collectively confirm the broad applicability and effectiveness of AMA and FastAMA as general, modular enhancements for attacking MLLMs.

## 4.5 ABLATION STUDY

To comprehensively analyze the effectiveness and efficiency of our proposed AMA and FastAMA modules, we conduct a series of ablation studies on InternVL model.

Table 3: Attack success rates (%) of different integration strategies for combining AMA/FastAMA with MF-Attack on InternVL model.

| Integration Strategy | VQA-Hard | VQA-Easy | Classification | Captioning |
|---|---|---|---|---|
| w/o AMA/FastAMA | 34.6 | 69.0 | 48.2 | 18.9 |
| Fixed $\lambda$ | 40.1 (+5.5) | 72.3 (+3.3) | 47.7 (-0.5) | 17.7 (-1.2) |
| Uncentainty Weighting | **53.2** (+18.6) | **79.7** (+10.7) | **51.0** (+2.8) | **19.4** (+0.5) |
| Disparity-Reduced Filter | 50.8 (+16.2) | 77.6 (+8.6) | 49.3 (+1.1) | 19.3 (+0.4) |

Table 4: Attack success rates (%) of different attention weighting strategies for AMA with MF-attack on InternVL model.

| Weighting Strategy | VQA-Hard | VQA-Easy | Classification | Captioning |
|---|---|---|---|---|
| Averaging | 50.8 | 75.3 | 50.2 | **19.5** |
| Edit-Distance | **53.4** | **79.1** | 50.8 | **19.5** |
| Semantic-BERT | 52.2 | 78.7 | **51.0** | 19.3 |

### 4.5.1 INTEGRATION STRATEGY

We first investigate the effectiveness of different ensemble strategies when integrating AMA/FastAMA with existing attack methods. As shown in Table 3, we compare three integration methods on MF-Attack: (1) using a fixed weighting coefficient $\lambda$; (2) applying an uncertainty-based adaptive weighting scheme (Uncentainty Weighting) Cipolla et al. (2018), where $\lambda$ is dynamically adjusted during the attack process based on the variance of task-specific losses, aiming to balance the original loss with the AMA objective; and (3) avoiding loss-level ensemble altogether by instead performing gradient-level fusion. Specifically, we adopt the Disparity-Reduced Filter (DRF) from AdaEA Chen et al. (2023), which computes the gradients of the two losses independently and filters out pixel-wise gradients with high directional inconsistency, thus improving attack stability. Table 3 reports the averaged results across both AMA and FastAMA variants, demonstrating that using a fixed $\lambda$ brings limited improvement, and in some cases even slightly degrades performance. In contrast, both Uncertainty Weighting and DRF lead to noticeable performance gains, with Uncertainty Weighting consistently outperforming DRF across all tasks. Therefore, we adopt Uncertainty Weighting as the default ensemble strategy for both AMA and FastAMA.

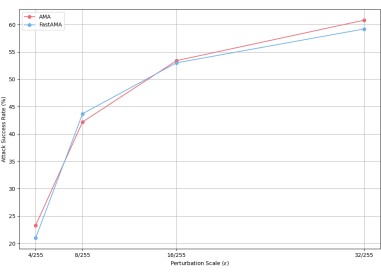 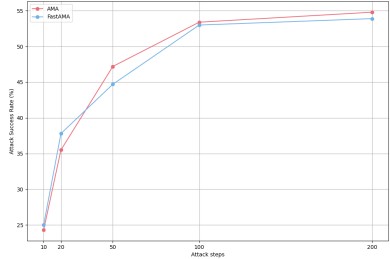

(a) Ablation on perturbation scale ($\epsilon$)   (b) Ablation on total steps of attack

Figure 3: Ablation experiment in different perturbation scale and attack steps settings, for AMA and FastAMA with MF-attack on InternVL model.

### 4.5.2 ATTENTION WEIGHTING STRATEGY

We further compared three strategies for setting attention weights in AMA: uniform averaging (assigning equal weights to all tokens), edit-distance-based weighting, and semantic weighting based on Sentence-BERT (SBERT) Reimers & Gurevych (2019) similarity. As shown in Table 4, for MF-Attack on InternVL model, the edit-distance-based strategy achieved the best performance, while SBERT weighting slightly underperformed.

We hypothesize that this is mainly because AMA targets fine-grained, token-level corrections between the model's prediction and the ground-truth output. Edit distance explicitly captures token-level misalignments, providing clearer, more precise guidance for AMA to focus attention on the most critical discrepancies. In contrast, SBERT generates smooth, global sentence embeddings, which are less sensitive to individual token errors. Consequently, its similarity scores may not provide sufficiently strong or localized signals to guide adversarial perturbations effectively.

Additionally, since our attack targets—whether the ground truth answers in the VQA-Hard task or the "Unknown" tokens used in other tasks following prior work—are typically very short words or phrases, this brevity likely further favors the effectiveness of edit-distance weighting over SBERT. Short target texts reduce the benefit of global semantic embeddings and emphasize the importance of token-level alignment. Therefore, we applied edit-distance as the weighting strategy for both AMA.

### 4.5.3 PERTURBATION SCALE AND STEPS

As shown in Figure 3a and Figure 3b, we evaluate the Attack Success Rate (ASR) on the VQA-Hard task targeting InternVL under different settings of perturbation scale and attack steps, with AMA/FastAMA integrated into MF-Attack.

For perturbation scale, we fix the number of steps to 100 and test four values: 4/255, 8/255, 16/255, and 32/255. As illustrated in Figure 3a, the ASR increases steadily as the scale enlarges. Even at 4/255, a non-trivial attack success can be observed, while 16/255 yields a significantly higher ASR. Considering the need to preserve human-perceivable image quality and to ensure fair comparison with prior work, we set the perturbation scale to 16/255 in our final setup.

For the number of attack steps, we fix the scale to 16/255. As shown in Figure 3b, the ASR improves with more steps and converges around 100 steps, beyond which the gains become marginal.

### 4.5.4 EFFICIENCY

We further compare the efficiency of AMA and FastAMA in terms of the average time required to generate each adversarial sample and the peak GPU memory usage, when integrated with MF-Attack on the VQA-Hard task using InternVL. As shown in Table 5, when the total steps of attack is set to 100, FastAMA significantly improves generation speed—achieving approximately a 6.9× acceleration compared to AMA—while also reducing peak memory usage by about 37%. Meanwhile, as demonstrated in Table 1 and Table 2, the attack success rate of FastAMA only slightly drops relative to AMA. These results highlight the practical necessity and efficiency advantages of FastAMA in real-world adversarial scenarios.

Table 5: Comparison of AMA and FastAMA on average time consumed per image, peak memory usage and attack success rate(%), with MF-attack on InternVL model.

| Method | Time consuming | Memory usage | ASR |
|--------|----------------|--------------|-----|
| AMA | 430.1s | 22942MB | 53.4 |
| FastAMA | 62.3s | 14443MB | 53.0 |

## 5 CONCLUSION

In this work, we propose Attention Misalignment Attack (AMA) and its efficient variant FastAMA, novel plug-and-play strategies for crafting adversarial examples against Multimodal Large Language Models (MLLMs). Unlike existing methods that primarily target model outputs, AMA explicitly manipulates the attention alignment between visual and textual tokens, enabling more effective disruption of the model's internal inference process. Through extensive experiments across multiple tasks—including image classification, captioning, and both standard and challenging VQA datasets—and on various MLLMs such as LLaVA, MiniGPT-4, and InternVL, we demonstrate that AMA consistently improves attack success rates across different attack baselines. Moreover, FastAMA achieves comparable effectiveness while significantly reducing computational cost. Our results highlight the importance of targeting the attention mechanism in MLLMs and provide a new perspective for future adversarial research in multimodal models.

ETHICS STATEMENT

This work adheres to the ICLR Code of Ethics. In this study, no human subjects or animal experimentation was involved. All datasets used were sourced in compliance with relevant usage guidelines, ensuring no violation of privacy. We have taken care to avoid any biases or discriminatory outcomes in our research process. No personally identifiable information was used, and no experiments were conducted that could raise privacy or security concerns. We are committed to maintaining transparency and integrity throughout the research process.

REPRODUCIBILITY STATEMENT

We have made every effort to ensure that the results presented in this paper are reproducible. The experimental setup, including attack steps, model configurations, and hardware details, is described in detail in the Sec. 4.2. We have also provided a full description of AMA/FastAMA in Sec. 3.2, to assist others in reproducing our experiments. Additionally, the dataset used in this work, such as TextVQA Singh et al. (2019), MS-COCO Lin et al. (2014), are publicly available, ensuring consistent and reproducible evaluation results. We believe these measures will enable other researchers to reproduce our work and further advance the field.

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

## LLM USAGE

Large Language Models (LLMs) were used to aid in the writing and polishing of the manuscript. Specifically, we used an LLM to assist in refining the language, improving readability, and ensuring clarity in various sections of the paper. The model helped with tasks such as sentence rephrasing, grammar checking, and enhancing the overall flow of the text.

It is important to note that the LLM was not involved in the ideation, research methodology, or experimental design. All research concepts, ideas, and analyses were developed and conducted by the authors. The contributions of the LLM were solely focused on improving the linguistic quality of the paper, with no involvement in the scientific content or data analysis.

The authors take full responsibility for the content of the manuscript, including any text generated or polished by the LLM. We have ensured that the LLM-generated text adheres to ethical guidelines and does not contribute to plagiarism or scientific misconduct.

