# OpenReview forum: "Attention Misalignment Attacks: Targeting Cross-Modal Attention in Multimodal Large Language Models for Adversarial Examples"
_ICLR.cc/2026/Conference — ICLR 2026 Conference Withdrawn Submission_

### Official Review · Reviewer_Qn2r · 2025-10-26

**Soundness:** 2
**Presentation:** 2
**Contribution:** 2
**Rating:** 2
**Confidence:** 4

**Summary:**

This paper proposes a method called Attention Misalignment Attack (AMA) to create adversarial examples for Multimodal Large Language Models (MLLMs). The core idea is to add a loss term to existing attack frameworks that maximizes the divergence of attention maps between a clean and a perturbed input. This is intended to disrupt the model's ability to ground its textual responses in relevant visual information. The authors also present a more efficient version, FastAMA, and test the method's ability to boost the performance of several existing attacks, particularly on Visual Question Answering (VQA) tasks.

**Strengths:**

Focused Investigation: The work provides a focused investigation into the role of attention maps as an adversarial attack surface, which is a plausible mechanism for disrupting MLLMs.

Practical Modularity: The method is designed as a "plug-and-play" module, which offers some practical utility for researchers looking to marginally improve existing attack pipelines.

Efficiency Variant: The development of FastAMA shows consideration for the computational overhead of adversarial attacks, which is a practical and necessary step.

**Weaknesses:**

Limited Novelty and Contribution: The central weakness of this paper is its limited novelty. The idea of attacking a model's internal representations is not new. Targeting attention maps is an incremental and unsurprising extension of this general strategy, rather than a novel paradigm. The method is ultimately an auxiliary loss function that provides a slight boost to existing techniques, which calls into question the significance of its contribution.

Overstated Claims: The paper frames this incremental improvement as a "novel framework," which seems to overstate the contribution. The reported performance gains are most pronounced in the specific "VQA-Hard" setting defined by the authors, but are less impressive on other tasks, suggesting the method's utility may be narrow.

Weaknesses in Evaluation: The evaluation feels one-sided. It focuses solely on attack success without considering the robustness or transferability of the generated examples in the presence of even simple defenses. The ablation studies, while present, are not entirely convincing; the justifications for certain design choices (e.g., weighting strategies) feel post-hoc and may not represent fundamental insights.

**Questions:**

Given that targeting internal representations is a well-established adversarial strategy, could you clarify what makes targeting attention maps fundamentally different or more insightful than, for example, targeting the output of intermediate feature layers?

The proposed method directly manipulates a core architectural component (the attention mechanism). Does this specificity have a negative impact on the transferability of the adversarial examples to different MLLM architectures? A successful attack that is not transferable has limited practical security implications.

---

### Official Review · Reviewer_r8rb · 2025-10-30

**Soundness:** 2
**Presentation:** 2
**Contribution:** 2
**Rating:** 4
**Confidence:** 4

**Summary:**

The paper focuses on adversarial attacks of MLLMs, and it proposes a mis-alignment adversarial attack to perturb MLLM hidden embeddings for generating better attack performance.

**Strengths:**

1. The paper focuses on an intriguing and important research question.
2. AMA achieves better performance compared to traditional attacks.

**Weaknesses:**

The paper mainly has problems in the following aspects:

1. Writing issues. The writing is unclear. Starting from Section 3, the authors do not define their threat model—is this a white-box or a black-box attack? In Section 2, what is A_l^{\prime} and how is it obtained, especially under different settings, e.g., untargeted vs. targeted attacks? Ambiguous notation also hurts readability: the formulas use w_l, while the text uses w_t. Are they the same thing? In addition, I don’t understand the motivation for using edit distance to measure token similarity—shouldn’t the similarity between the target word and the generated word be measured semantically?

2. Insufficient related-work discussion. Injecting perturbations into intermediate model features is not a new topic; it has been widely studied in prior work [1,2]. The authors should discuss how this paper differs from those studies to help readers better understand the technical contribution.

[1] A self-supervised approach for adversarial robustness. CVPR 2020

[2] FDA: feature disruptive attack. ICCV 2019

3. Experimental issues. The authors only compare against LLaVA, MiniGPT-4, and InternVL-2.5. The paper lacks experiments on more advanced MLLMs such as Qwen.25-VL, and LLaVA-1.5. In addition, comparisons with existing adversarial attack methods for MLLMs [3,4] are missing; these should be included to demonstrate the method’s effectiveness.

[3] A Frustratingly Simple Yet Highly Effective Attack Baseline: Over 90% Success Rate Against the Strong Black-box Models of GPT-4.5/4o/o1. NeurIPS 2025

[4] M-Attack-V2: Pushing the Frontier of Black-Box LVLM Attacks via Fine-Grained Detail Targeting. arXiv 2025

**Questions:**

Please refer to the weakness section.

---

### Official Review · Reviewer_Jvth · 2025-10-31

**Soundness:** 3
**Presentation:** 3
**Contribution:** 3
**Rating:** 4
**Confidence:** 4

**Summary:**

The proposed Attention Misalignment Attack (AMA) method primarily addresses the problem of adversarial attacks on MLLMs in the context of fine-grained understanding tasks. AMA operates by directly manipulating the model's internal cross-modal attention maps during the decoding process. This forces the model to attend to irrelevant image regions, thereby misguiding its fine-grained comprehension. Furthermore, the authors introduce a lightweight variant, FastAMA, which approximates the attack by using a single forward pass and extracting the self-attention map from the final layer, significantly enhancing efficiency. Experiments conducted on LLaVA, MiniGPT-4, and InternVL models demonstrate that AMA/FastAMA can significantly boost the attack success rate of existing methods on tasks like VQA (especially VQA-Hard).

**Strengths:**

1. Novel Attack Perspective

The paper accurately identifies the reason for the failure of existing attacks on fine-grained tasks and, in response, shifts the attack target from the model's final output to its internal attention mechanism. This approach directly targets the core mechanism of MLLM visual reasoning, is logically sound, and the use of KL divergence to measure attention misalignment is a reasonable and effective choice.

2. Efficient Variant (FastAMA)

The authors clearly recognize the computational bottleneck of the AMA method (i.e., extracting attention at each decoding step is too slow) and proactively propose FastAMA. It requires only a single forward pass. As shown in Table 5, FastAMA achieves an approximately 7x speedup and a 37% reduction in memory usage, with only a marginal drop in attack success rate. This significantly enhances the practical utility of the method.

**Weaknesses:**

1. The paper fails to discuss the transferability of adversarial examples generated from one white-box model to another. Consequently, the method's effectiveness in black-box scenarios is highly questionable.

2. The work is limited to targeted attacks. The "attention misalignment" concept seems inherently bound to guiding the model's attention toward a specific target pattern. It is unclear how this loss function would be defined or optimized in a simpler, untargeted attack setting (i.e., where the goal is simply to force any incorrect answer).

3. The ablation study is limited. It primarily discusses the settings for the lambda parameter in the integration strategy and the w_l in attention weighting. It fails to explore the individual contributions of AMA's different components (e.g., the weighting strategy itself, or the integration with existing attack methods) to the overall attack success. A more comprehensive ablation study would be more persuasive.

**Questions:**

1. As mentioned in Weakness 1, the method relies heavily on white-box access. Have the authors tested the transferability of adversarial examples generated by AMA/FastAMA in a black-box setting? For example, can samples generated on LLaVA successfully attack InternVL? What is the potential of this method against practical black-box models?

2. In Section 4.2, the paper states that for VQA-Easy, Classification, and Captioning tasks, the attack target is uniformly set to the generic token "Unknown". However, in Section 4.5.2, "edit distance" is chosen as the optimal strategy for computing token weights w_l, citing its ability to provide precise token-level guidance. How is a meaningful w_l weight, capable of guiding gradient optimization, calculated when the target is such a single, generic token like "Unknown"?

3. For FastAMA, the authors chose to use the self-attention map from the decoder's final layer. Did the authors experiment with using other layers or a fusion of multiple layers? Why is the final layer the optimal choice?

---

### Official Review · Reviewer_UGpH · 2025-11-01

**Soundness:** 3
**Presentation:** 3
**Contribution:** 2
**Rating:** 4
**Confidence:** 4

**Summary:**

This paper proposes Attention Misalignment Attack (AMA) and its efficient version FastAMA, plug-and-play adversarial methods designed to disrupt cross-modal attention in MLLMs. By optimizing the divergence between target and adversarial attention pattern, the approach aims to misguide fine-grained visual understanding. The method is evaluated across multiple models and attack baselines, demonstrating significant improvements in attack success rates.

**Strengths:**

- Experimental results: The method consistently improves attack success rates across models and tasks.

- Efficiency: FastAMA offers a compelling speedup with minimal performance drop, making it suitable for resource-constrained scenarios.

- Compatibility: The plug-and-play design allows easy integration with existing attack frameworks, enhancing reproducibility and applicability.

**Weaknesses:**

- Limited attack scenario: The method is strictly limited to white-box settings, as it requires access to intermediate attention maps from the MLLM decoder. This restricts its practicality in real-world applications (e.g., cloud-based MLLM APIs) where model internals are inaccessible.

- Novelty concerns: While targeting attention mechanisms is valuable, the core idea of using intermediate features (e.g., attention maps) to enhance adversarial attacks has been widely explored in prior works (for instance, Kim et al. (2024).


- Limited insight and broader impact: The experiments focus heavily on attack success rates but lack deeper analysis of how attention misalignment affects model robustness and interpretability. For example, does AMA reduce the transferability of adversarial examples compared to baseline attacks?

- Technical clarifications needed, e.g., the size of victim models.

**Questions:**

Please see the weakness part.

---

### Note · Authors · 2025-11-21

I have read and agree with the venue's withdrawal policy on behalf of myself and my co-authors.